# Spreading Senescent Cells’ Burden and Emerging Therapeutic Targets for Frailty

**DOI:** 10.3390/cells12182287

**Published:** 2023-09-15

**Authors:** Serena Marcozzi, Giorgia Bigossi, Maria Elisa Giuliani, Giovanni Lai, Robertina Giacconi, Francesco Piacenza, Marco Malavolta

**Affiliations:** 1Advanced Technology Center for Aging Research and Geriatric Mouse Clinic, IRCCS INRCA, 60121 Ancona, Italy; s.marcozzi@inrca.it (S.M.); g.bigossi@inrca.it (G.B.); m.giuliani@inrca.it (M.E.G.); r.giacconi@inrca.it (R.G.); f.piacenza@inrca.it (F.P.); 2Scientific Direction, IRCCS INRCA, 60124 Ancona, Italy

**Keywords:** aging, frailty, cellular senescence, immunosenescence, microbiome, virome

## Abstract

The spreading of senescent cells’ burden holds profound implications for frailty, prompting the exploration of novel therapeutic targets. In this perspective review, we delve into the intricate mechanisms underlying senescent cell spreading, its implications for frailty, and its therapeutic development. We have focused our attention on the emerging age-related biological factors, such as microbiome and virome alterations, elucidating their significant contribution to the loss of control over the accumulation rate of senescent cells, particularly affecting key frailty domains, the musculoskeletal system and cerebral functions. We believe that gaining an understanding of these mechanisms could not only aid in elucidating the involvement of cellular senescence in frailty but also offer diverse therapeutic possibilities, potentially advancing the future development of tailored interventions for these highly diverse patients.

## 1. Introduction

A substantial body of literature suggests that the excessive accumulation of senescent cells is likely to play a role in the development and progression of several age-related diseases [1]. The concept that cellular senescence can spread through paracrine and systemic signals to neighboring cells and distant tissues is perhaps the most exciting aspect of the kinetics of senescent cell accumulation [2,3,4,5,6,7,8,9,10]. The inflammatory components of the senescence-associated secretory phenotype (SASP) are not the only players in this phenomenon. Most recent evidence has demonstrated that the SASP includes extracellular vesicles (EVs) through which senescent cells exert central effector functions in the local environment [11,12,13]. Senescent cells secrete EVs with a distinctive, albeit incompletely characterized, content of miRNA, proteins, and DNA that can spread senescence in surrounding and even distant tissues [14,15,16], thereby promoting further inflammation and catastrophic consequences for the organism [7]. Since frailty is defined as a condition characterized by increased vulnerability to internal and external stressors [17,18], we can argue that the uncontrolled spreading of senescence in response to stress could contribute to the accumulation of deficits that characterize frailty. While the accumulation of senescent cells in every organ or tissue is likely to contribute to frailty, musculoskeletal and cerebral health are the two aspects that are most prominently represented in the instrumental tools to assess frailty [19,20,21,22,23]. Likely, these cells exert their detrimental effects not only when localized specifically in the muscle (e.g., satellite senescent cells) or the brain (e.g., senescent glial cells or senescent-like neurons) but also in other tissues, where they can impact the brain and muscles through secreted factors.

A recent study has investigated the kinetics and factors driving the spreading of senescence into neighboring tissues (a phenomenon also called the “bystander effect”) [4]. The authors proposed a model for the accumulation rate of senescent cells based on the sum of the accumulation rate due to replicative exhaustion and damage and the accumulation rate due to the bystander effect, subtracted by the rate of immunodegradation of senescent cells.

The significant bystander effect, coupled with the reduced efficiency of the immune system in clearing senescent cells, has the potential to make organisms more vulnerable. This vulnerability arises from the ability of these senescent cells, formed in response to specific types of damage, to rapidly and systematically spread throughout the organism. This process may explain why frailty can emerge as a result of a singular pathology or why the severity of frailty can differ among individuals.

In particular, we focused on microbiome dysregulation, as recent evidence has suggested its direct impact on the accumulation of senescent cells and its robust association with frailty. Alterations in both the viral and bacterial components of the human gut microbiota were observed in association with aging, and the pharmacological modulation of these components may affect frailty and cellular senescence.

In the following sections, we will describe some established and innovative factors that should deserve particular attention in frailty because of their potential to promote a rapid, systemic spreading of senescent cells in response to stress (Figure 1).

## 2. Failure of “Senescence Immunosurveillance”

The immune system plays a crucial role in controlling the spreading of senescence and the net accumulation of senescent cells. The mechanisms through which the immune system detects and eliminates senescent cells, known as “senescence immunosurveillance”, display close similarities to those proposed for eliminating (pre-)malignant cells. Senescent cells can expose specific immune ligands on their surface to mediate the recognition by immune cells, including Natural Killer (NK) cells, monocytes/macrophages, and T cells, which are recruited by some components of the SASP [5,26]. With advancing age, the immune system appears to maintain a permanent state of mild activation and a reduced ability to respond to new antigens, a phenomenon collectively termed immunosenescence [27,28,29,30,31].

Aging is also commonly accompanied by low-grade chronic inflammation, termed “inflammaging”. This low-grade inflammatory process may originate from the response to persistent pathogens or endogenous stressors, as well as the accumulation of senescent cells and their associated SASP [32,33,34]. Inflammaging and immunosenescence are two sides of the same coin and are suggested to be among the causes of most age-related diseases. However, this negative interpretation has been challenged by an increasing number of immune-gerontologists, as these changes can be viewed as adaptive or remodeling phenomena that may be needed for extended survival/longevity. This topic has already been extensively addressed by specific reviews [28,33,34], so it will not be a focal point of discussion in our work.

Some aspects of immunosenescence have already been proposed among the factors contributing to frailty [32]; however, recently, evidence has been provided that highlights a direct link between failing immunosurveillance and accelerated aging [35]. Moreover, in specific biological contexts, some senescent cells have been shown to promote immune suppression through paracrine signals and MMP-dependent shedding of the NK cell receptor (NKG2D) [36], a mechanism of paramount importance for the immunosurveillance of senescent cells [37]. The failure of immunosurveillance is considered a significant aspect of the growth of certain types of cancers [38]. In contrast, the lack of proliferation of senescent cells could lead to underestimating the phenomenon in aging and frailty [39,40]. The spreading of senescence mediated by circulating extracellular vesicles could lead to a profound re-evaluation of this phenomenon in aging and frailty [11,12,13]. For instance, senescent cells contribute significantly to the plasma extracellular vesicle pool in old mice, potentially affecting the function of cells throughout the body [15].

Hence, the failure of senescence immunosurveillance may be considered a critical aspect for potential spreading of the senescent cell burden. Importantly, the emerging evidence supports that the composition of the intestinal microbiota strongly influences senescence [41,42,43,44,45,46,47] and cancer immunosurveillance [48,49,50], offering new therapeutic perspectives for frailty.

## 3. Alteration of Microbiome and Virome

Alteration of the microbiome, which encompasses both the bacterial and viral components (virome), is another emerging mechanism that, under specific stressing conditions, may drive an increased net accumulation rate of senescent cells [51].

### 3.1. The Microbiome

The gut microbiota and their metabolites are associated with multiple musculoskeletal health deficits, including sarcopenia, osteoporosis, osteoarthritis, and rheumatoid arthritis [52]. Similarly, disturbances along the brain–gut–microbiota axis seem to significantly contribute to the pathogenesis of neurodegenerative disorders [53].

A bidirectional relationship between the microbiome and cellular senescence has been proposed [51]. Secretory metabolites of the microbiome can directly impact cellular senescence (e.g., in intestinal cells), while the accumulating senescent cells may contribute through the SASP to altered immune functions.

It has been documented that the gut microbiota profiles of frail older people are different from those of non-frail, older individuals [54,55,56,57]. Moreover, a multitude of stressors and interventions (such as psychological stress, circadian disruption, sleep deprivation, environmental temperature, environmental pathogens, drugs, toxicants, pollutants, noise, physical activity, and diet) alter the composition, function, and metabolic activity of the gut microbiota [58,59]. A growing body of evidence indicates that alterations in the microbiota are linked to age-related diseases and negative health outcomes, while some interventions with prebiotics and probiotics have displayed preventive and pro-longevity effects [60,61,62,63,64,65]. Hence, it is not surprising that microbiome disturbance is now considered among the hallmarks of aging [66].

Specific gut microbiota-dependent metabolites have been recently shown to promote the appearance of senescent cells and the SASP in some tissues. One of these metabolites (trimethylamine-N-oxide) increases with aging in circulation and seems to be able to promote senescence in the vascular endothelium [67]. Other gut microbial metabolites, such as lipoteichoic acid and deoxycholic acid, have been shown to induce senescence in hepatic stellate cells, promote the production of SASP, and suppress immunosurveillance in the liver [41,42]. Other bioactive metabolites such as lactate, isocitrate, citrate, and malate are known to accumulate with chronological age in both human and mice and have been implicated in various aspects of aging and age-related diseases [68]. Some of these bioactive metabolites, such as lactate, are not only part of the secreted array of molecules produced by senescent cells but are also produced by the gut microbiota [45]. Others, such as citrate, isocitrate, and malate, are likely related to gut microbiota through indirect mechanisms that involve the modulation of cellular metabolism [43,44].

Concurrently, the extract of the bacterium *Sphingomonas hydrophobicum* can delay skin senescence in a reconstructed skin model, as demonstrated by the reduced activation of SA-β-gal activity and cell cycle inhibitors p21 and p16Ink4a [46]. Similarly, another study showed that the application of probiotic bacteria, such as *Lactobacillus fermentum* isolated from human fecal matter, demonstrated a beneficial effect in suppressing stress-induced permanent cellular senescence: it mitigates multiple senescence markers that are characteristic of preadipocytes, such as DNA damage response and cell cycle inhibition signaling, cellular hypertrophy, SA-β-gal activity, the activation of SASP, and the Akt/mTOR pathway [47].

The microbiome can also influence anti-tumor immunosurveillance through direct and indirect mechanisms, mediating, at least in part, the increased cancer risk of sedentary and other unhealthy lifestyles [48]. Similar immunosuppressive mechanisms have been proposed to explain the association between colorectal cancer and alterations of the oral bacterial and fungal microbiome [49]. Significantly, the oral microbiota varies with different comorbidities, degrees of frailty, and, obviously, the presence of teeth [50].

### 3.2. The Virome

While the microbiome is now widely studied in aging mice and humans, much less knowledge is available about the role of the vast numbers of different viruses, collectively termed the virome, hosted in all mammals [69,70]. Part of the virome is increasingly recognized as an essential component of the microbiome, and virome changes have been appreciated as indicators of immune status [71,72]. Human virome includes eukaryotic viruses, endogenous retroviruses, and bacteriophages [73].

#### 3.2.1. Persistent and Latent Pathogenic Viruses

Whether components of the virome can contribute to the spreading of cellular senescence, either through direct or indirect immune-mediated mechanisms, is a current research challenge. However, most viruses, especially latent viruses, have the capacity to reactivate in response to various types of stress. In this context, the marked vulnerability to stress that characterizes frailty appears more than a simple coincidence.

Evidence that certain viruses are involved in telomere shortening, mitochondrial dysfunction, oxidative stress, DNA damage, and other molecular mechanisms leading to cellular senescence has been extensively provided [74,75,76]. Moreover, cellular senescence has also been described as an anti-viral mechanism [77,78,79,80,81,82]; thus, it is unsurprising that many viruses, including SARS-CoV-2, have been shown to directly induce cellular senescence [80,83,84]. Recently, a systematic network-based analysis of the human and viral protein interactomes produced a list of the top viral candidates predicted to influence human aging; among these, the influenza A virus (subtype H1N1) has emerged as the leading candidate, primarily due to its genetic, encoded potential to interact with cellular senescence [85].

While there is mounting evidence that the gut microbiota profiles of frail older individuals differ from those of non-frail older people, research regarding the virome is still in its early stages. A recent study [86] has shown the differences between frail and non-frail individuals in the abundance of a specific circulating virome component, torquetenovirus (TTV), a commensal human *annellovirus* representing the most abundant component of human virome. High TTV viremia has also been associated with an increased risk for all-cause mortality in an Italian elderly population study [87].

Human immunodeficiency virus (HIV-1) is known to cause premature aging and to dramatically increase the degree of frailty in chronically infected patients [88]. Interestingly, HIV-1 glycoprotein gp120 and transactivator of transcription (Tat) can stimulate the release of endothelial microvesicles, which promote inflammation, oxidative stress, and cell senescence [89]. HIV-1 also induces a senescence-like phenotype in human microglia. The transfer of supernatants from infected to naïve microglia cultures resulted in senescence induction and the release of pro-inflammatory factors [76].

Albeit there are contradictory results [90]; some studies have found that cytomegalovirus (CMV) infection contributes to frailty syndrome and mortality risk in the elderly through mechanisms that promote immunosenescence [91,92]. Similarly, women who are seropositive for herpes simplex virus types 1 (HSV-1) and 2 (HSV-2) but not varicella-zoster virus (VZV) and Epstein–Barr virus (EBV) have been shown to display a higher risk of 3-year incident frailty [93].

The virome can influence anti-tumor immunosurveillance through both direct and indirect mechanisms. For example, certain viruses, such as human papillomavirus (HPV) and EBV, can affect the tumor microenvironment by upregulating immunosuppressive pathways [94]. In this context, it has been observed that certain viruses possess microRNAs that are recognized for their ability to inhibit specific components of the immune system [95,96,97], impair senescence immune surveillance, and promote an accelerated spreading of senescent cells. The herpes simplex virus (HSV) has been observed to interfere with dendritic cell (DC) viability and function, consequently impairing immune surveillance [98,99]. Additionally, TTV was found to increase in individuals with Down syndrome, a human model of accelerated immunosenescence, while it was reduced in centenarian offspring [100].

#### 3.2.2. Endogenous Retroviruses and Bacteriophages

A special note is warranted regarding the role of a subpopulations of viruses, namely, retroviruses and bacteriophages.

It has been suggested that the genomic DNA demethylation associated with aging may induce the upregulation of human endogenous retroviruses (HERVs), “ancient viruses”, constituting approximately 8% of the human genome. The expression of HERV-H, HERV-K, and HERV-W families increases in the peripheral blood of subjects over 60 years old [101], and they undergo epigenetic alterations repeatedly observed in the context of organismal and cellular senescence in humans and other species [102]. The activation of HERVs was also observed in the organs of aged primates and mice, as well as in human tissues and serum from older adults [103]. In human senescent cells, HERV-K can produce retrovirus-like particles to induce paracrine senescence, whereas the repression of HERVs alleviates cellular senescence and tissue degeneration [103]. However, there is no specific study addressing HERVs in frailty, and there is no evidence of an association between the HERV-K family and immunosenescence markers [104].

Regarding bacteriophages, a significant and abundant component of the human gut virome, there are still no studies focused on frailty or cellular senescence. A recent study showed that the human intestinal virome changes with age, with a more diverse virome in centenarians compared to gut viromes of younger adults (>18 year) and older individuals (>60 year) [105].

This component of the virome does not display pathogenic effects under normal conditions, but studies highlight a delicate balance between viral symbiosis and pathogenesis during chronic infections and immunodepression [106]. Bacteriophages are known to strongly affect the function and composition of the bacteriome [107]. They may have a direct impact on inflammation in the case of specific diseases, such as Crohn’s disease [108]. The circulating virome of patients with cardiovascular disease, a condition linked to cellular senescence by epidemiological and experimental evidence, was enriched with bacteriophages compared to healthy controls [109]. Further studies on the modulation of the virome in experimental models of frailty may be helpful to advance this field of research.

## 4. Therapeutic Strategies to Target Senescence in Frailty

Senescent cell spreading can be triggered by internal or external stressors. Gaining insight into the underlying mechanisms responsible for the loss of control over senescent cell spreading holds significant importance in frailty, which is characterized by extreme vulnerability to stressors. This understanding is pivotal not only for deciphering the varying extents of vulnerability in the elderly with frailty but also for refining therapeutic approaches.

### 4.1. Senolytics and Senomorphics

To date, the overwhelming evidence of the beneficial effects of senescent cell elimination has led to the identification of a new class of drugs, called senotherapeutics, as a possible new approach to ameliorating age-related diseases. Senotherapeutic molecules could be classified in two categories [110]: senolytics, which selectively eliminate senescent cells, and senomorphics, which suppress the markers of senescence or components of the SASP [111,112,113].

Senolytics are a heterogeneous class of compounds that exert their function by targeting the molecular mechanisms involved in the anti-apoptotic and pro-survival processes that lead to the induction of the senescent phenotype. Among the most important are the dasatinib plus quercetin cocktail, navitoclax, ABT-737, fisetin, Proxofim, Geldanamycin, Tanespimycin, Foxo4-DRI, Panobinostat, Azithromycin, Roxithromycin, EF24, and UBX0101, and other drugs based on therapeutic nanoparticles and gene therapy are currently being developed [114]. These compounds target a multitude of survival pathways (tyrosine kinase receptors, BCL-2, PI3K, autophagy, p53, HSP90, and OXR1), which are differently activated in different senescent cells, making each one selective for a subset of these cells depending on the type of senescence and the tissue of origin.

The number of senomorphic drugs has also expanded in these last years, including Simvastatin, Kaempferol, Apigenin, Ruxolitinib, Metformin, Rapamycin, Loperamide, NDGA, Cortisol, KU-60019, NDGA, and SB203580. Most of these compounds display anti-inflammatory activity, which appears to be related to their capacity to inhibit the major pathways involved in the production of the SASP, such as NF-κB, p38MAPK, JAK, and TOR [114,115]. Interestingly, also non-pharmacological treatments can have senostatic effects, such as dietary restrictions [116] and, maybe, physical exercise [117,118]. All these therapeutic opportunities are of paramount relevance in the perspective of translation into the clinic treatment of patients with frailty. However, each treatment may have side effects, and the limited clinical evidence to date on senotherapeutics suggests moving toward a cautious, patient-tailored approach.

### 4.2. Probiotics, Prebiotics, and Antivirals

We have identified at least two other potential therapeutic targets, alterations of the microbiome and virome, that may be implicated in senescence immunosurveillance and thus in the rapid spread of senescence.

Understanding how these factors increase the senescence burden will be challenging, but several interventions to improve the gut microbiota’s homeostasis have been successfully tested in pre-clinical and clinical experimental studies. For instance, the oral administration of the genus *Akkermansia* in mice (a mouse probiotic identified from “rejuvenation” studies) improved senescence-related phenotypes in the intestinal system, the frailty index (FI), cognitive activity, muscle function, and extended health span [119]. However, in both mice and humans, there is still insufficient information about the appropriate dosages and administration periods of probiotics and prebiotics [120]. Nonetheless, some clinical studies have shown positive health outcomes in older people with frailty [63]. Administration of inulin and fructooligosaccharides (commercially available prebiotics) in the older adults resulted in a significant reduction in FI [121] and improved exhaustion and handgrip strength, two criteria of the physical frailty phenotype (PP) [122].

There is also evidence that pharmacological targeting of certain aspects of the virome may impact frailty and cellular senescence. HIV-positive patients undergoing antiretroviral therapy not only have a higher risk of non-infective comorbidities and multimorbidity but also of geriatric syndromes and frailty [123]. Emerging studies in pre-clinical models also support the notion that antiretroviral drugs used to manage HIV may be able to suppress senescence and age-associated inflammation [124].

## 5. Conclusions and Future Directions

This review highlights the significant role of the microbiome on the initiation and propagation of cellular senescence, identifying alterations in the microbiome and virome as potential therapeutic targets for frailty and age-associated diseases.

It is worth noting that studies on aging or frailty often face challenges in determining causality due to the complexity of the biological processes involved. Thus, it cannot be excluded that the changes in the microbiome could be consequential and possibly late events in frailty rather than causal. In spite of this, they can still be targets for pharmacological interventions aimed at combating frailty.

Hence, combining senotherapeutics with probiotics/prebiotics and/or antiretroviral agents has indicated a potential toward frailty with a high potential for rapid translation into humans (Figure 2).

Likely, the choice of senotherapeutics for use in combined therapy will be a challenge. Senolytics are the most likely candidates, but the administration of senolytics based on oral flavonoids, such as the dasatinib plus quercetin cocktail and fisetin, should be implemented carefully, as they are likely to directly interfere with the microbiota composition [125,126]. Other solutions, based on the subcutaneous or intravenous injection of highly selective senolytics (e.g., genetic-based therapies) or the use of specific transgenic mouse models (i.e., p16-3MR and INK-ATTAC models), seem more suitable for planning the initial pre-clinical experiments for this kind of combined therapy.

Other interventions based on physical exercise, nutrition, lifestyle changes, adjustment of poly-pharmacy, or even anti-inflammatory treatments have shown beneficial effects in frail patients [127,128,129,130].

For example, it has been suggested that the effects of caloric restriction might be partially attributed to a reduction in the number of senescent cells [131]. However, it is also known that caloric restriction induces a profound remodeling of the gut microbiome, and it is reasonable to hypothesize a correlation between the two phenomena [62,132,133].

Some of these studies have the potential for synergic activity with senotherapeutics and have been shown to modulate both cellular senescence [134] and the microbiome [59].

Further work should be addressed to understand whether changes in the microbiome associated with frailty can affect senescence immunosurveillance in frail individuals.

As pointed out by others [135,136], identifying a circulating biomarker as a surrogate for tissue accumulation of senescent cells would greatly benefit this area of research.

In conclusion, the rapid and excessive accumulation of senescent cells resulting from changes in the microbiome/virome and the failure of senescence immunosurveillance are likely to occur in frail individuals under various types of stress. These findings provide a basis for developing combination therapies that target senescent cell accumulation at different levels, such as senolytics (downstream action) and modulators of the microbiome/virome (upstream action).

However, it is important to acknowledge that a substantial portion of the supporting evidence originated from murine models. The limitations of these strategies could be related to the fact that some interventions in mice may not be translated to humans and that targeting senescent cells may be more challenging than previously thought, as some senescent cells may play important roles in the structural and metabolic functions of different tissues [137,138].

It is also crucial to bear in mind that the diet and microbiome of humans and mice differ significantly (e.g., mice are typically fed with a standardized diet; certain bacterial species present in the human gut may be absent in mice and vice versa; they differ for the anatomy and physiology of the digestive systems, etc.), which may lead to differing responses to potential treatments between the two species [139,140].

Careful consideration and validation are necessary to translate research outcomes on the intestinal microbiome from murine models to humans, taking potential pitfalls into account.

## Figures and Tables

**Figure 1 cells-12-02287-f001:**
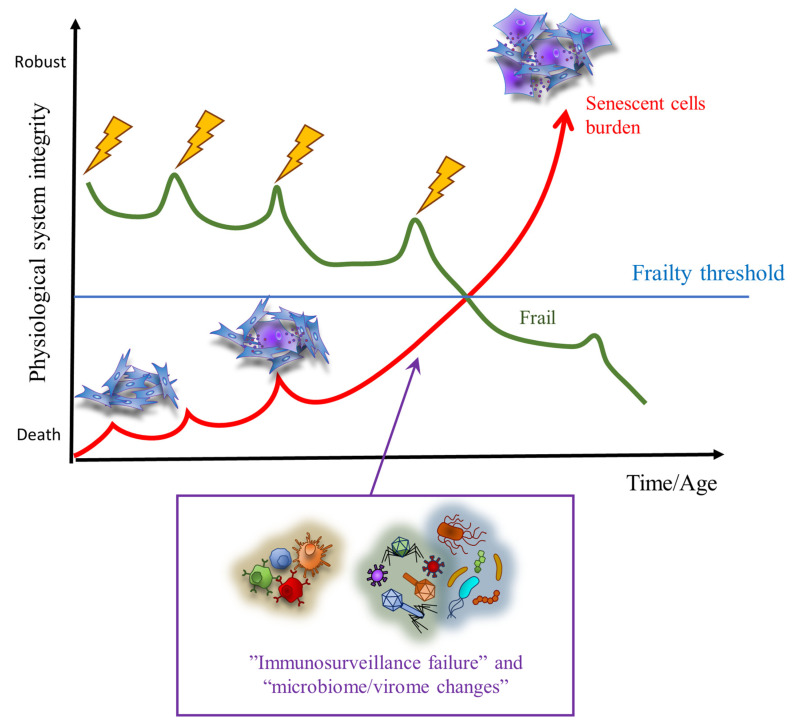
Relationship between stressors and accumulation of senescent cells resulting from failure of senescence immunosurveillance and changes in the microbiome/virome. The image is adapted and redrawn from Fried et al., 2021 [24] and Xue et al., 2019 [25].

**Figure 2 cells-12-02287-f002:**
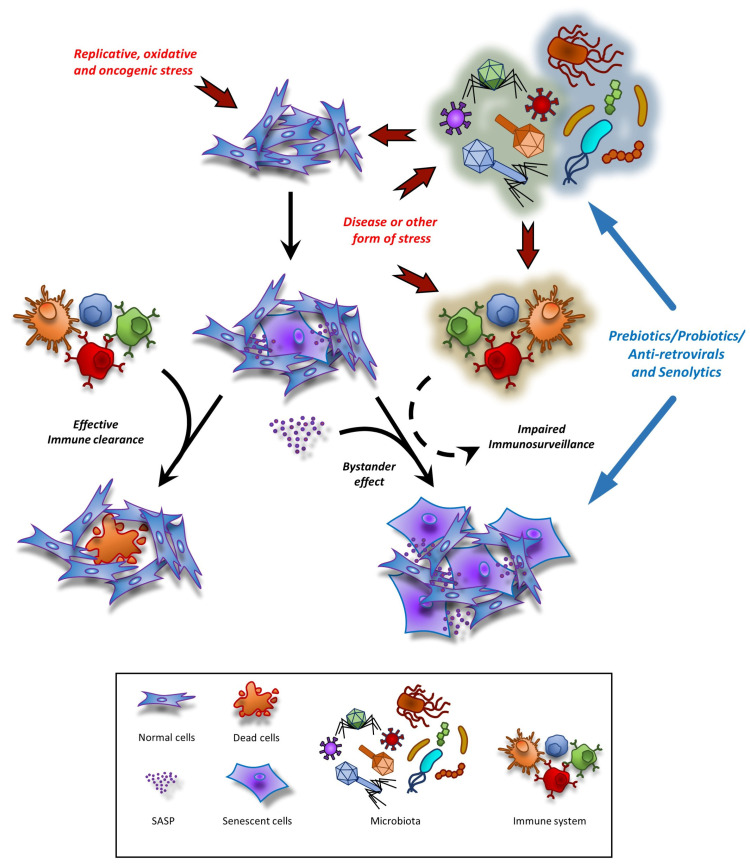
Overview of mechanisms that can contribute to the rapid spreading of senescent cells in aging. Damage (e.g., replicative, oxidative, and oncogenic), impaired immunosurveillance, and persistent SASP secretion synergistically increase senescent cells’ burden in organs and tissues. Disease or other forms of stress (such as diet, environmental factors, or aging) can induce structural and functional changes in the microbiota that, in turn, can directly affect tissues or the immune system’s ability to respond to stress, such as causing systemic inflammatory aggravation. Combined therapies targeting senescent cells at different levels, such as senolytics (downstream action) and modulators of the microbiome/virome (upstream action), could display synergic beneficial effects in the therapeutic and preventive approaches for frailty.

## Data Availability

No data were used for the research described in the article.

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
