# Peer review of "Spreading Senescent Cells’ Burden and Emerging Therapeutic Targets for Frailty"

_cells, 2023, doi:10.3390/cells12182287_

Round 1

Reviewer 1 Report

Please see document attached

Moderate editing of English language required

Author Response

Spreading of Senescent Cells Burden and Emerging Therapeutic Targets for Frailty Suggestion: major revisions

  1. Please select either perspective or review for the type of article- not both

We selected perspective as the type of article.

  1. Please use a more scientific word for ‘spreading’ in your title – increased/ proliferation/metastasis etc to make the title more professional

We understand the reviewer concern regarding the word "spreading." However, it's worth noting that the term "spreading" is commonly used in the context of cellular senescence within the scientific literature (PMID: 34302996; PMID: 22321662; PMID: 35241831; etc etc.). While we respect the recommendation to consider alternative terms like "increased," "proliferation," or "metastasis," we believe that these options may not accurately convey the senescent state and the specific nuances of our perspective article. Hence, we believe that “spreading” remains the most appropriate choice to describe the phenomenon.

  1. Please provide recent evidence/ references for lines 31-32, for every sentence between line 84-88, lines 92-95, 98-99,

We have now added the references [11-13] at line 33, [38] at line 102, [39,40] at line 103, [11-13] at line 105, [41-47] and [48-50] at line 111, and [51] at line 117, accordingly.

  1. On what basis has figure 1 been prepared? Was there data to suggest the lines drawn in the figure? What do each of the peaks in the green and red lines indicate? What is the scale on the x and y axes? At the moment – the figure lacks publication quality and appears made up with lines drawn without any given evidence. This figure is not needed as you clearly explain it in text, please remove it as it is not adding to your article. If you want to keep it, please change it considerably and cite the reference (if the figure was inspired) in the figure legend.

We apologize for the missing information. The image has been adapted and redrawn based on Figure 4 from Fried et al., 2021 [23] and Xue et al., [24]. In comparison to the original, in our figure, we have included the concept of 'senescent cells burden. We have now correctly added the reference in the figure caption.

  1. Please use Biorender for creating publication quality figures, especially figure 2. You can use the free version to create up to 5 figures

We have carefully considered the option of using Biorender to create high-quality figures, especially for Figure 2. However, we have seen that the free version of Biorender allows to prepare and download 5 images at 72 dpi only for presentations, websites, dissertations etc, but they cannot be used for Journal publications without a paid account (https://help.biorender.com/en/articles/5898558-where-can-i-use-my-illustrations). Moreover, we already used this type of figure in a previous publication [PMID: 36496992]. We think that the high-resolution color figure, which ensured the required quality for publication, was not available in the pdf or docx version revised by the reviewer. We have now uploaded the HR version that we deem can satisfy the Journal and the reviewer criteria, but we are still open to considering any further suggestions or specific modifications to further enhance Figure 2.

  1. Please use ‘with advancing age’ instead of ‘during aging’ throughout your document

We changed this sentence accordingly.

  1. Reference 19 is quite old considering the field of immunosenescence. Please cite 3-4 references talking about immunosenescence in the last 3-5 years. Here are four relevant references that you may find useful: Immunosenescence and Inflamm-Aging as Two Sides of the Same Coin: Friends or Foes? Front. Immunol. 2018; b. Immunosenescence and Its Hallmarks: How to Oppose Aging Strategically? A Review of Potential Options for Therapeutic Intervention. Front. Immunol. 2019; c. Immunosenescence in aging: Between immune cells depletion and cytokines up-regulation. Clin. Mol. Allergy 2017; d. Aging, bone marrow and next-generation sequencing (NGS): Recent advances and future perspectives. Int. J. Mol. Sci. 2021

We incorporated the references recommended by the reviewer (27, 28, 29, 30 and 31) at line 84.

  1. Please talk about virome and microbiome in the introduction in short to better connect the flow of the article. Currently, section 3 comes out of nowhere as these topics were not mentioned even in brief (4-5lines) in the introduction. Mentioining them only in the abstract is not enough

We thank the reviewer for the valuable suggestion. We have now integrated a sentence in the introduction about virome and microbiome at line 58-62.

  1. Line 140-141 should be added in the end of the paper under ‘future directions’

The sentence from line 140-141 has been moved to the section 6, ‘conclusions and future directions’ at line 340-341.

  1. Please separate the microbiome and virome parts in section 3 as sub-sections 3a and 3b respectively as you discuss them separately

We have now divided the section 3 in subsection “3.1. The microbiome” and “3.2. The virome”.

  1. For the statement supported by reference 54, is there any other evidence suggesting senescence to be an anti-viral mechanism? You have used only one evidence and that too is really old. Anything more robust suggesting the same in the last 3-5 years? Please include 2-3 references to support this statement, especially considering that viruses are usually deemed to be quite harmful and negatively impact the immune system.

We thank the reviewer for this suggestion. We have now added four additional and more recent references, accordingly.

  1. The virome part of section 3 lacks flow and connection to your current article and is not able to provide a purpose to your work. Please rearrange the virome part with as much clarity as your micribiome section has

Based on the reviewer suggestion, we have carefully revised the virome section of our article to enhance its flow and connection to the overall narrative of the work. We believe that these revisions have significantly improved the clarity and purpose of the virome portion, aligning it more closely with the microbiome section. We hope the reviewer will appreciate these changes.

  1. You haven’t discussed any therapeutic strategies that currently exist for senescent cells – like senolytics, senomorphics etc until section 4. Please include a short paragraph of the current strategies currently available before you start discussing the suggested targets for therapeutic approaches. Please retitle section 4 as therapeutic target for senescence and discuss the points mentioned here. Then in the last 2-3 paragraphs, title that as section 5 Conclusions and future directions and include them respectively

We thank the reviewer for the feedback and suggestions. We have incorporated the recommendations into our manuscript. We renamed Section 4 to “Therapeutic Strategies to Target Senescence in Frailty” and included, within it, two subsections: “4.1. Senolytics and Senomorphics” where we discuss the current therapeutic strategies that exist for senescent cells and “4.2. Probiotics, Prebiotics, and Antivirals”. Additionally, we have created a new section titled “5. Conclusions and Future Directions”.

  1. Please rephrase line 222-223 in fewer, more scientific language

We have decided to remove the sentence in question.

  1. Line 237: FI – write in full

We included the explanation of the acronym FI as suggested.

  1. Line 239; please replace ‘is a promising treatment’ to ‘has indicated potential towards....’ S it is yet to be proven

We changed this sentence, accordingly.

  1. If you are suggesting therapeutic targets, please discuss for what pathways/mechanisms do you think these may be used for. Currently this essential aspect of proposing therapeutic target is completely missing in the article. If needed, discuss this in a separate section or sub-section with figures/tables. How do you envision that these two targets that you have suggested may be used for therapy and what would be the potential mechanism of action? This insight will help strengthen your manuscript further

We have added a new paragraph titled “4. Therapeutic Strategies to Target Senescence in Frailty" where we have provided a more detailed explanation and introduced new insights to provide a more complete explanation of how both targets, cellular senescence and the microbiome, can be harnessed to reduce inflammation and enhance the phenotype of frail individuals. We hope these revisions address the reviewer’s concerns and contribute to the overall clarity of the manuscript.

  1. English proofread is needed – please check

We have had the paper proofread by a professional language editor to ensure its linguistic accuracy and readability.

Reviewer 2 Report

Topic selection is of high interest and the article summarizes its field well. While there is suprisingly high emphasis on viruses, another contradictory field is rather neglected. The authors emphasize how immune suppression develops along with age. However, the scientific community also accepts and supports the concept of inflamm-aging, which proposes an elevated immune steady-state with age. It would be most welcome to discuss these opposing themes, pros and cons for both sides.

Author Response

Topic selection is of high interest and the article summarizes its field well. While there is suprisingly high emphasis on viruses, another contradictory field is rather neglected. The authors emphasize how immune suppression develops along with age. However, the scientific community also accepts and supports the concept of inflamm-aging, which proposes an elevated immune steady-state with age. It would be most welcome to discuss these opposing themes, pros and cons for both sides.

We thank the reviewer for this thoughtful and insightful feedback on our manuscript. We are pleased to hear that you find our topic selection of high interest.

We also acknowledge that this is indeed an intriguing area of research but also a complex topic. The interplay between inflammaging and immunosuppression has been subject of significant discussion and has been thoroughly examined in several dedicated reviews (e.g. PMID: 29375577; PMID: 32949594; PMID: 37291596). We believe, as indicated also by others, that inflammaging and immunosuppression are interconnected phenomena, representing two sides of the same coin. This means that there is a mutual interaction between the inflammaging-producing factors inducing immunosenescence and the immunosenescence-producing factors which contribute to the maintenance of the inflammaging. We have now incorporated this concept into our manuscript in Chapter 2 at line 85-94.

Reviewer 3 Report

This review is interesting in that unlike many others it focuses on the role of the microbiome and the virome in the induction and spreading of cellular senescence in frailty. The review is somewhat speculative but is well written and provocative. Unfortunately, the review is not within the expertise of this reviewer.

A few points to consider:

1. Throughout the authors should consider the issue of cause and effect as the changes in microbiota could be consequential and possibly late events in frailty rather than causal. The effects of viruses and infectious agents are likely causal.  However, but what do they have to do with ageing?

2. There is no mention of bioactive metabolites such as lactate (Martinez-Outschoornet al Cell Cycle 2011) and citrate (Birkenfeld et al Cell Metabolism 2011, Fan et al J .Gerontology A 2021; Aging Cell 2021, Willmes et al JCI Insight 2021), which are part of the secreted array of molecules produced by senescent cells (Martinez-Outschoorn et al Cell Cycle 2011; James et al J. Proteome Res. 2015) and which accumulate with chronological age in both humans (Menni et al Int. J. Epidemiol. 2013) and mice (Varshavi et al. Front Mol Biosci 2018; Yue et al. Biomolecules 2022).

Other metabolites such as isocitrate and malate are associated with frailty (Marron et al Metabolites 2019) health (Cheng et al Nature Comm. 2015); Yeri et al J. Gerontology A 2019) and cardiovascular disease (Cheng et al 2015) in humans . The deletion of senescent cells in mouse models ameliorates and slows age-related diseases but does not prevent them and so other factors, including metabolites, may be relevant. It might be worth mentioning how the issues raised by the authors influence ageing via plasma metabolites.

3. Cellular senescence are both regulated by energy consumption in parallel (Fontana et al Aging Cell 2018) and so the above points are highly relevant to any review on ageing and senescence. 

4. A large amount of the evidence that showing that senescent cells contribute to ageing and age-related disease is derived from genetic and pharmacological depletion of senescent cells in mice and so some comment on whether this applies to humans in a ‘limitations of current studies’ might be advisable for balance.

For example, if the authors examine the human IL-6 data in reference 6 (Jeon et al Nature metab.2022) the accumulation of plasma IL-6 with age in humans is not that great and is not statistically significant whereas the mouse data is much more convincing. This is particularly pertinent following the recent disclosures that the mouse diet in many (if not most) experiments may be deficient in NAD+ precursors which do not decline in human ageing (Sun et al EMBO J 2020). SASP proteins are also very low in human disease and variable  (see JL Kirkland clinical trial results in eBiomedicine 2019 to 2023). I think some caution is merited here despite the justified excitement about the potential of senescence-related diagnostics and therapeutics.

5. In Section 3 the authors mention metabolites but do not list them except trimethylamine-N-oxide put them into the context of senescence. What does trimethylamine-N-oxide do and how does it compare to other age- and senescence-associated metabolites. Reference 38 implies it exerts its effects through oxidative stress and despite many comments in the affirmative there is very little evidence to support this in ageing, especially in humans (see Campisi et al Nature 2019) and human plasma metabolomics indicates that the ROS-sensitive TCA cycle continues to function even in the very elderly (Mota-Martorell et al Free Radic Biol Med. (2021) and frail (Marron et al Metabolites 2019).

6.  In the conclusions, the authors could point out the differences between mouse and human diet (see also above) and their microbiota.

7. References

The original references demonstrating the local spread of senescence in vitro (Nelson et al Aging Cell 2012) and both in vitro and in vivo (Acosta et al Nature Cell Biology 2013) should be cited.

Author Response

This review is interesting in that unlike many others it focuses on the role of the microbiome and the virome in the induction and spreading of cellular senescence in frailty. The review is somewhat speculative but is well written and provocative. Unfortunately, the review is not within the expertise of this reviewer.

A few points to consider:

  1. Throughout the authors should consider the issue of cause and effect as the changes in microbiota could be consequential and possibly late events in frailty rather than causal. The effects of viruses and infectious agents are likely causal. However, but what do they have to do with ageing?

We thank the reviewer for bringing up this intersting point. While not all elderly is or will be frail, frailty is an extreme consequence of the normal ageing process (Hoogendijk, Lancet 2019). Hence, it is quite complicated, if not impossible to disentangle “frailty specific causes” from aging processes. It is worth noting that studies on aging or frailty often face challenges in determining causality due to the complexity of the involved biological processes. Thus, it cannot be excluded that the changes in microbiome could be consequential and possibly late events in frailty rather than causal. In spite of this, they can still be targets for pharmacological interventions aimed at combating frailty. We discussed this point in the Section 5 at line 315-319.

  1. There is no mention of bioactive metabolites such as lactate (Martinez-Outschoornet al Cell Cycle 2011) and citrate (Birkenfeld et al Cell Metabolism 2011, Fan et al J .Gerontology A 2021; Aging Cell 2021, Willmes et al JCI Insight 2021), which are part of the secreted array of molecules produced by senescent cells (Martinez-Outschoorn et al Cell Cycle 2011; James et al J. Proteome Res. 2015) and which accumulate with chronological age in both humans (Menni et al Int. J. Epidemiol. 2013) and mice (Varshavi et al. Front Mol Biosci 2018; Yue et al. Biomolecules 2022). Other metabolites such as isocitrate and malate are associated with frailty (Marron et al Metabolites 2019) health (Cheng et al Nature Comm. 2015); Yeri et al J. Gerontology A 2019) and cardiovascular disease (Cheng et al 2015) in humans . The deletion of senescent cells in mouse models ameliorates and slows age-related diseases but does not prevent them and so other factors, including metabolites, may be relevant. It might be worth mentioning how the issues raised by the authors influence ageing via plasma metabolites.

We are grateful to the reviewer for this important suggestion, which we have briefly discussed in the chapter 3.1 at line 143-149.

  1. Cellular senescence are both regulated by energy consumption in parallel (Fontana et al Aging Cell 2018) and so the above points are highly relevant to any review on ageing and senescence.

We have taken into account the reviewer suggestion and we have now added the concept of caloric restriction in the discussion at line 334-337.

  1. A large amount of the evidence that showing that senescent cells contribute to ageing and age-related disease is derived from genetic and pharmacological depletion of senescent cells in mice and so some comment on whether this applies to humans in a ‘limitations of current studies’ might be advisable for balance. For example, if the authors examine the human IL-6 data in reference 6 (Jeon et al Nature metab.2022) the accumulation of plasma IL-6 with age in humans is not that great and is not statistically significant whereas the mouse data is much more convincing. This is particularly pertinent following the recent disclosures that the mouse diet in many (if not most) experiments may be deficient in NAD+ precursors which do not decline in human ageing (Sun et al EMBO J 2020). SASP proteins are also very low in human disease and variable (see JL Kirkland clinical trial results in eBiomedicine 2019 to 2023). I think some caution is merited here despite the justified excitement about the potential of senescence- related diagnostics and therapeutics.

We thank the reviewer for these suggestions. We fully agree with the reviewer's insights. Additionally, we believe that the limitations extend beyond this aspect to the broader question of whether all senescent cells should be eliminated or only specific ones. We have briefly introduced these considerations in the “Conclusion and future directions” section of the manuscript at line 350-355.

  1. In Section 3 the authors mention metabolites but do not list them except trimethylamine- N-oxide put them into the context of senescence. What does trimethylamine-N-oxide do and how does it compare to other age- and senescence-associated metabolites. Reference 38 implies it exerts its effects through oxidative stress and despite many comments in the affirmative there is very little evidence to support this in ageing, especially in humans (see Campisi et al Nature 2019) and human plasma metabolomics indicates that the ROS- sensitive TCA cycle continues to function even in the very elderly (Mota-Martorell et al Free Radic Biol Med. (2021) and frail (Marron et al Metabolites 2019).

We focused on trimethylamine- N-oxide (TMAO) because most of its production in the body relies on the gut microbiome, and because it has attracted attention in recent years due to its potential association with heart disease and other health conditions. We also mentioned two other metabolites (lipoteichoic acid and deoxycholic acid) since it has been reported that these metabolites, like TMAO, can induce senescence in certain cells. We have now added in section 3.1 at lines 143-149 other metabolites that are produced by senescent cells and may be indirectly/directly related to microbiome alterations.

  1. In the conclusions, the authors could point out the differences between mouse and human diet (see also above) and their microbiota.

Unfortunately, in this perspective article, we cannot address all the differences between mouse and human diet in their full extent. However, we have added a sentence at line 356-360 to acknowledge our awareness of this limitation. Certainly, there are several differences between the diets and gut microbiota of mice (rodents) and humans that can have significant implications when studying mouse models and translating findings to humans.

  1. References. The original references demonstrating the local spread of senescence in vitro (Nelson et al Aging Cell 2012) and both in vitro and in vivo (Acosta et al Nature Cell Biology 2013) should be cited.

We have added the references in the text at line 29.

Round 2

Reviewer 1 Report

Minor editing of English language required

Author Response

We have done a full spell check and corrected various minor typos (including one in Figure 2). Many thanks for this further suggestion.